

# Effects of chemical fungicides combined with plant resistance inducers against *Bipolaris sorokiniana* in turfgrass

Filiz Ünal[1], İlker Kurbetli[2], Yeşim Eğerci[3] and Aysun Cavusoglu[4]

[1] Faculty of Agriculture, Eskişehir Osmangazi University, Eskişehir, Türkiye
[2] Plant Health Department, Batı Akdeniz Agricultural Research Institute, Antalya, Türkiye
[3] Department of Cereal Diseases, Plant Protection Research Institute, İzmir, Türkiye
[4] Faculty of Agriculture, Kocaeli University, Kocaeli, Türkiye

## ABSTRACT

One of the most common and causative diseases problems disease in turfgrass areas in Türkiye is leaf blight, root and crown rot caused by *Bipolaris sorokiniana*. The fungus causes great damage especially in parks, refuges, and golf courses, and needs to be managed. This study aimed to determine some activators alone or in combination with effective fungicides at different doses against *B. sorokiniana*, to reduce the use of fungicides in the control of the disease. In the study, the effects of combinations of different doses of some fungicides with *Lactobacillus acidophilus*, *Arthrobacter* sp. and a harpin protein determined to be effective in *in vitro* studies were investigated in greenhouse and field conditions in two different provinces. The highest effect was obtained in the combination of (*Lactobacillus acidophilus*-(Azoxystrobin+Difenocazole)), which was used at the recommended dose (94.70% effect) and the recommended dose of Azoxystrobin+Difenocazole (92.57% effect). The (*Lactobacillus acidophilus*-(Azoxystrobin+Difenocazole 1st subdose)) application, in which a lower dose of fungicide was used, ranked 4th with an 89.50% effect. When *Lactobacillus acidophilus*, *Arthrobacter* sp. and Harpin were used alone, they were found to be 82.53%, 72.17%, and 66.63% effective against the disease, respectively. As a result, environmentally friendly low-dose fungicide, and activator combination (*Lactobacillus acidophilus*-(Azoxystrobin+Difenocazole 1. subdose)), and (*Arthrobacter* sp.-(Epoxiconazole+Pyraclostrobin 1st sub-dose)) applications were found to be promising in the control of *B. sorokiniana* in turfgrass areas. When considered only as an activator application, the 82.53% effect value obtained in *Lactobacillus acidophilus* applications was found to be promising for the disease. It can be recommended as a disease suppressant, and at the same time environmentally friendly application.

Corresponding author
Filiz Ünal, filiz.unal@ogu.edu.tr

## INTRODUCTION

Day by day turfgrass areas such as parks, gardens, sports areas, refuges, and golf courses are increasing all over the world as a result of the development of approaches based on modern urbanism and lifestyle (*Chawla et al., 2018*). Turfgrass areas that require continuous irrigation have increased, leading to disease-related problems. Especially

harmful soil-borne and leaf fungi severely damage turfgrass areas and managements of the diseases cause major economic and environmental problems (*Chawla et al., 2018*). *Bipolaris* sp., which was previously called previously *Helminthosporium* is a common pathogen of gramineae as well as cool climate turfgrass plants (*Manamgoda et al., 2014*). *B. sorokiniana* (Sacc.) Shoemaker (teleomorph *Cochliobolus sativus* (Ito and Kuribayashi) Drechs. Ex Dastur) is the most common and virulent member of this group (*Kumar et al., 2002*; *Smiley, Dernoeden & Clarke, 2005*). *B. sorokiniana* causes intense brown necrosis on turfgrass leaves. The damage of the fungus, which is the invasion of the leaf sheath and crown, called 'melting out' is more damaging to the turfgrass and can lead to the death of the entire plant. In such plants, purple lesions can cover the entire leaf and dry out the leaves. The fungus can invade the sheath, crown, rhizomes, and roots. Eventually, these plants die and turn brown or straw-colored. Severe melting-out can result in irregular patches of dead turf. Damaged lawns often appear "thin" or uneven (*Smiley, Dernoeden & Clarke, 2005*). The disease is a common and virulent leaf fungus in the turfgrass areas in Türkiye (*Yilmaz & Boyraz, 2007*).

Although the Kentucky bluegrass variety is known to be resistant to the disease, in areas where a wide spectrum of diseases occur in the same location each year, the use of resistant varieties or cultural measures may be insufficient. In these cases, fungicide application is necessary to suppress the development of the disease to tolerable levels. The most effective fungicide group against the disease is the QoI group (strobilurins) fungicides. The application timing is critical for satisfactory fungicide performance. In cases of severe infection, a single fungicide application is not effective in suppressing the disease; therefore, repeated fungicide applications are necessary (*Smiley, Dernoeden & Clarke, 2005*; *Latin, 2024*).

In cases of severe infection, especially in golf courses, stadiums, refuges and recreational areas where the incidence of this disease is high, a single fungicide application is not effective in suppressing the disease; therefore, repeated fungicide applications are necessary (*Smiley, Dernoeden & Clarke, 2005*; *Latin, 2024*). In addition to high costs, these practices pollute the environment and groundwater resources and cause resistance to fungicides (*Balci & Gedikli, 2012*; *Clarke et al., 2024*). The use of fungicides with the same active ingredient every year also causes resistance over time (*Delen, 2008*). For this reason, it is very important to reduce the amount and frequency of use of chemicals that are inevitably used in the fight. It is of great importance to include biological control studies to reduce the negative effects of chemical control.

A comprehensive study has been carried out to control turfgrass diseases and it has been determined that DMI, phenylamide and QoI group fungicides are highly effective against some turfgrass diseases, but these fungicides carry a high risk of resistance (*Nobutaka et al., 2006*; *Vincelli & Munshaw, 2017*). For this reason, it is recommended to use these fungicides in combination with other biological preparations or use them sequentially (*Nobutaka et al., 2006*).

Microorganisms that cause activation of plant immune responses are of great interest in agriculture as potential biocontrol agents as they can reduce pesticide use (*Kurokawa et al., 2021*). The most striking ones in this regard are the usage of certain microorganisms and

organic substances in disease control. Some of these microorganisms or microorganism-derived compounds activate plant resistance mechanisms, some of which protect plants from diseases when produced during competition for food and place struggle with antagonists, and some of which are biological activators that trigger plant resistance and provide effective diseases control results (*Bishnoi & Payyavula, 2004*; *Tezcan & Akbudak, 2009*).

Lactic acid bacteria such as *L. acidophilus* and *L. paracasei* during the fermentative growth period, produce metabolites with activating properties that optimize vital activities such as respiration, photosynthesis, protein and carbohydrate mechanisms and enzyme systems in plants (*Yu, Leveau & Marco, 2020*; *Forwood et al., 2022*; *Yang et al., 2024*). As plant growth regulators and fertilizers, these metabolites increase the quality of the grass by inducing strong growth in the root system of the grass. They also increase the density, texture, and color hereby highlighting grass quality (*Gaggia et al., 2013*; *Bosi et al., 2023*).

Harpin protein, which activates the natural defense mechanism of the plant, is a protein secreted by some plant pathogenic bacteria such as *Pseudomonas syringae* pv. *syringae*, and *Erwinia amylovora* (*Reboutier et al., 2007*). Harpin proteins have been preferred as an alternative to fungicides for years due to their low toxicity and low residue. Since harpin has no direct effect on pathogens, the risk of resistance development is also low (*Delen, 2008*). In Türkiye, commercial plant activators containing harpin protein are recommended officially as a single application in grasses at 2–3 leaves period. Harpin protein is recommended for grass in EPA (US Environmental Protection Agency Office) pesticide programs (*United States Environmental Protection Agency, 2023*).

*Arthrobacter* is a bacterial genus of interest to researchers working in agriculture, medicine, industry, and the environment because of its ability to activates the plant resistance mechanism, decomposes pesticides, fixes nitrogen, produces beneficial enzymes, sewage treatment (*Velázquez-Becerra et al., 2013*; *Fu et al., 2014*; *Kurokawa et al., 2021*).

Several biological control studies have determined that the antagonists of *Trichoderma harzianum* and some other fungi are effective against *B. sorokiniana* in rye and wheat leaves (*Salehpour et al., 2005*; *Singh et al., 2018*). Some strains of *Pseudomonas* spp., *Lysobacter enzymogenes* C3, and *Stenotrophomonas maltophilia* C3 bacteria have also been reported to have antagonistic effects against *B. sorokiniana* (*Giesler & Yuen, 1998*; *Kilic-Ekici & Yuen, 2004*). With this approach, the negative effects of chemical pesticides on the environment are reduced, the risk of resistance to fungicides that may occur with excessive and frequent use of pesticides decreases, and the effectiveness of disease control increases thanks to plant activators.

As in all agricultural fields, there is an obligation to develop an effective, ecologically friendly, and economically profitable control strategy against diseases in turf areas. For this purpose, the use of activators, fungicides and their combinations are included in this study.

## MATERIALS AND METHODS

### Determination of the *in vitro* activity of fungicides

In this study, virulent (98%) (Fig. 1). *B. sorokiniana* isolates that isolated from turfgrass areas in Bursa province, in Türkiye were used. Firstly, *in vitro* studies were

conducted to determine the efficacy of the fungicides against the virulent *B. sorokiniana* isolate before studies under greenhouse and area conditions. The effectiveness of the fungicides (Epoxiconazole+Pyraclostrobin, Metconazole, Chlorotholanil, Trifloxystrobin, Azoxystrobin+Difenocazole, Chlorotholanil+Thiophonate-methyl) against mycelial growth and spore germination of the fungus was determined. İn addition, the most effective fungicide doses and the sensitivity levels of the pathogen to the fungicides were determined. Dose ranges of 0 (control), 1, 3, 10, 30, and 100 μg/ml effective substance (E.S) for fungicides with non-specific effect site (Chlorotholanil, Chlorotholanil+Thiophonate-methyl) and 0 (control), 0.01, 0.03, 0.1, 0.3, 1, 3, 10, 30 μg/ml for fungicides with special effect site (Epoxiconazole+Pyraclostrobin, Metconazole, Trifloxystrobin, Azoxystrobin+Difenocazole) were used. Stock solutions were prepared for higher doses from which dilutions were made to obtain the desired fungicide doses of 10,000, 1,000 and 100 ppm E.S doses (*Shah et al., 2006*). Sterile distilled water was used for dilutions. To obtain the desired dose from the stock solutions, the required amount of fungicide solution was added to the sterilized and cooled medium. Then, equal amounts of the media containing or not containing the desired fungicide doses (control) were poured into sterile petri dishes and left to solidify for a while. Agar discs with a diameter of four mm, taken from the developing hyphae tips of the colonies of the agent, were cultured in PDA (Potato dextrose agar) medium and incubated for 4–5 days in an incubator set at 24 ± 2 °C. Experiments were designed with three replications. The colony diameter of isolates was measured and recorded at the end of the incubation period. The efficacy of fungicides was demonstrated according to the 50% effective dose (ED50) and the lowest dose (MIC) values, based on the diameter measurement values of mycelial growth after inoculation (*Eğerci & Kinay-Teksür, 2018*). ED50 values were found by applying percent improvement values to the control log-probit paper (*Georgopoulos, 1982*; *Beever, Laracy & Park, 1989*). The average mycelial growth rate of each isolate was calculated for each fungicide treatment and compared to the control. $ED_{50}$ which is the effective fungicide concentration that inhibits micellar growth by 50%, was calculated by estimating the reduction in growth as a percentage of the control. The logarithm of this figure was used to calculate the linear regression as follows; $\log_{10}$ (fungicide concentration) $= a.\log_{10}$ (% reduction in growth) $+ b$, Where; $a$ = slope and $b$ = intercept. The $ED_{50}$ was calculated using the following equation; $EC_{50} = 10^{\wedge([\log_{10}(50)-b]/a)}$. To determine the minimum inhibitory concentration (MIC), the lowest concentration of fungicide doses that prevented the germination of the isolate was examined. For this, the colonies were examined under a microscope at the end of incubation. The dose at which spore germination did not occur was calculated as the MIC value of that fungicide (*Delen, Yıldız & Maraite, 1984*; *Mair et al., 2016*).

### Efficacy of fungicide, plant activator, and their combinations against *B. sorokiniana* under greenhouse conditions

In the greenhouse studies, the effects of the most effective two fungicides against *B. sorokiniana* in Petri dish experiments and three activators (*L. acidophilus*, *Arthrobacter* sp. and Harpin Protein) were investigated. Applications were made at the doses and times given on their labels (Table 1). A turfgrass mix including different grass varieties (*Festuca*

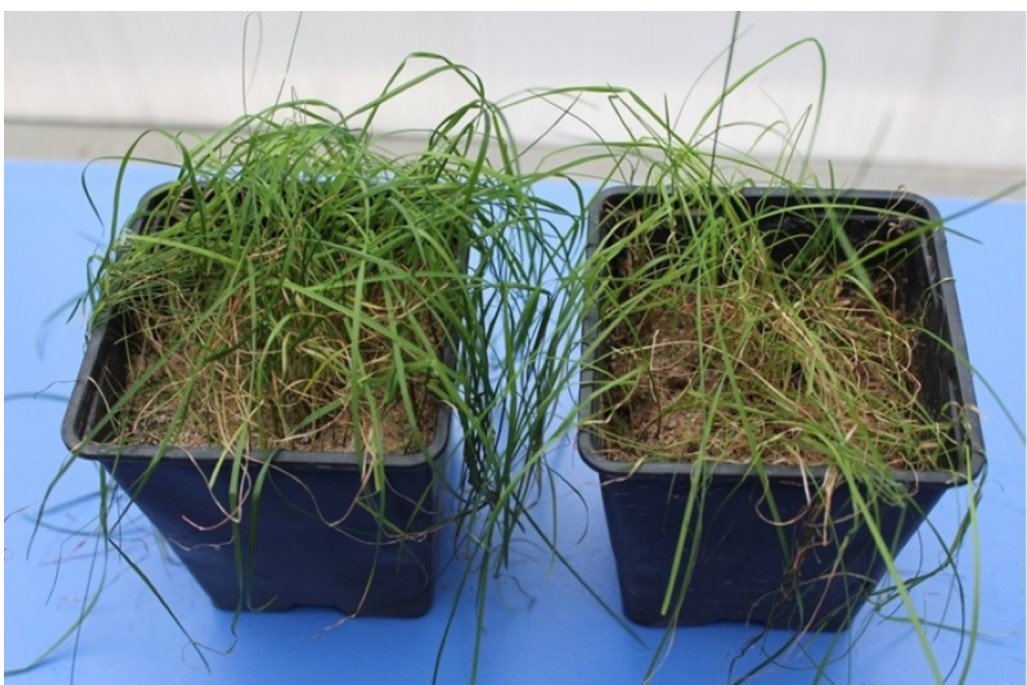

**Figure 1  Images from pathogenicity studies of *Bipolaris sorokiniana* on turfgrass.**

**Table 1  Application doses with fungicides and activators used in greenhouse trials.**

| Active ingredient | Application doses |
|---|---|
| Epoxiconazole+Pyraclostrobin | 200 mL/da, 100 mL/da, 50 mL/da |
| Azoxystrobin+Difenocazole | 80 mL/100 mL water, 40 mL/100 mL water, 20 mL/100 mL water |
| *Lactobacillus acidophilus* | 75 mL/da |
| Harpin Protein | 5 g/da |
| *Arthrobacter* sp. | 50 mL/2 L water (with 2.5 liters of water per m$^2$) |

spp., *Lolium* spp., *Agrostis* spp., *Poa* spp., and *Cynodon* spp.) these are commonly used in Türkiye was used in the experiment.

Greenhouse trials were conducted in three parts. In the first part (fungicide application only); the effectiveness of the two most effective fungicides in Petri experiments was assessed at their recommended dose and two subdoses. In the second part (activator application only), the efficacy of the three activators when used as a single application was determined. The third part (sequential application of activators and fungicides), was carried out as activator application first, and then fungicide applications at different doses. Experiments were carried out in three replications.

Fungal spores were obtained by adding sterile water containing 1% Tween 20 to cultures of *B. sorokiniana* isolate grown on PDA in Petri dishes and scraping the surface using a spatula. Then, the spore concentration was adjusted to $1 \times 10^5$ spores/ml using a Thoma counting chamber. In greenhouse studies, a mixture of sterile garden soil (autoclaved at

121 °C for 45 min), burnt farm manure and fine sand (2:1:1) were filled into $10 \times 10 \times 11$ size and 0.8 L volume sterile plastic pots. Turfgrass seeds were sown two cm deep into the mixed substrate; each pot had 35 seeds. The plants were kept in a greenhouse at $25 \pm 2$ °C for 15 days. The studies were carried out with three replications with each pot serving as a replicate. The fungicide, activator and activator-fungicide (consecutive) prepared at the prescribed doses were sprayed on the 15-day-old plants using a hand sprayer until the plants were thoroughly wet. The spore suspension was sprayed on plants one day after the fungicide application in only fungicide applications, and seven days after the activator application in only the activator application, until the plants were thoroughly wet. In the sequential use of activator-fungicide, the activator was applied first, a week later the fungicides and then, after one day the fungal pathogen was applied. After inoculation, the pots with plants were placed in moistened polyethylene bags to maintain a 95–100% relative humidity for 72 h (*Albayrak, 1991*; *Koo et al., 2003*). Treatments to control pots were made with sterile water. Assessments were made 25 days after the trials were completed according to the 0–5 scale below (*Adlakha, Wilcoxson & Raychaudhuri, 1984*). 0 = no spots, 1 = Less than 5% necrotic spots, 2 = Necrotic spots and yellowing covering 6–20% of the leaf, 3 = Significant necrotic spots and yellowing covering 21–40% of the leaf, 4 = Significant necrotic spots and yellowing covering 41–60% of the leaf, 5 = Distinct necrotic spots covering more than 60% of the leaf.

## Efficacy of fungicide, plant activator and their combinations against *B. sorokiniana* in area conditions

The two most effective fungicide + activator combinations in the greenhouse trials were applied in the field. Field trials were established by artificial inoculation in two locations with different climates (Ankara province Agricultural Protection Central Research Institute and Antalya province Western Mediterranean Agricultural Research Institute research fields). Ankara has a slightly more continental climate, while Antalya has a warmer and more humid climate. In the study, turfgrass varieties mix commonly used in Türkiye (*Festuca* spp., *Lolium* spp., *Agrostis* spp., *Poa* spp. and *Cynodon* spp.) that similarly were used in the greenhouse study. Turfgrass seeds were sown at 50 g/m². After sowing, previously prepared soil mortar with a thickness was sprinkled on the seeds as a cover not exceeding 0.5–1 cm (1/3 sand + 1/3 garden soil + 1/3 burnt farm manure) and then pressed with a pressing tool. Irrigation was done immediately after sowing. For proper germination of seeds, daily irrigations were done. The fungal pathogen was inoculated as in the greenhouse trials with three replications, two programs, besides negative and positive control; with 12 blocks for each application according to the randomized blocks trial designed. A 0.5 m safety strip was left between the parcels in the application area. The area of each plot was 4 m² (*Tosun & Turan, 2011*). The plots were divided with the help of rope and stakes. Evaluations were determined as percent area, considering the coverage rate of the disease in the trial plots using a scale 0–5, where 0 = no disease; 1 = 1–5 percent; 2 = 6–10 percent; 3 = 11–25 percent; 4 = 26–50 percent; 5 = >50% of plot symptomatic (*Gleason, Batzer & Johnson, 2011*). Treatments were applied to plants just as in the greenhouse experiment. Applications in control plots were made with sterile water.

In the experiments, the re-isolations were made from diseased plants. The identification of the developing fungus was made by examining the conidia and conidiophore structures under a light microscope (*Ellis, 1971*).

In the greenhouse and area assays, disease severity values were calculated using the 0–5 scales and the Townsend–Heuberger formula (*Townsend & Heuberger, 1943*). The obtained disease severity values were applied to the Abbott formula (*Abbott, 1925*) to determine the effectiveness (%) of the fungicides (*Stevic, Vuksa & Elezonovic, 2010*; *Torguet et al., 2022*). The re-isolations were made from the infected leaves. This study was carried out in a completely randomized design (CRD) comprising of three replicates. Statistical analysis was carried out using SPSS 16.0 (SPSS Inc., Chicago, IL, USA). A two-way ANOVA was done to determine differences among treatment groups for the parameters. The Duncan's Multiple Range Test was also done to determine if differences between individual treatments were significant ($P \leq 0,05$).

Townsend and Heuberger formula: Disease severity (%) = $[\sum (n.V)/Z.N] \times 100$

$n$: Number of samples hitting different disease degrees on the scale

$V$: Scale value

$Z$: Highest scale value

$N$: Total number of samples observed

Abbotts formula (*Abbott, 1925*) given as % efficacy = $[(X - Y)/X] \times 100$,

$X$—disease severity in the control treatment; $Y$—disease severity in fungicide treatment.

## RESULTS

### *In vitro* determination of the activity of fungicides

The effects of different doses of six fungicides on *B. sorokiniana* isolate were investigated in *in vitro* studies and Table 2 shows the $ED_{50}$ and MIC ($\mu$g/ml) values.

Epoxiconazole+Pyraclostrobin, Azoxystrobin+Difenocazole and Trifloxystrobin preparations significantly inhibited the mycelial growth of the pathogen (Table 2). The $ED_{50}$ value for Epoxiconazole+Pyraclostrobin was 0.04 $\mu$g/ml, 0.41 $\mu$g/ml for Azoxystrobin+Difenocazole and 0.44 $\mu$g/ml for Trifloxystrobin. Chlorotholanil+Thiophonate-methyl ($ED_{50}$:>100 $\mu$g/ml) and Chlorotholanil ($ED_{50}$:68 $\mu$g/ml) combinations did not inhibit the mycelial growth of *B. sorokinianan* even at high doses.

Chlorotholanil+ Thiophonate-methyl and Chlorotholanil preparations had the lowest inhibitory effects (MIC) on *B. sorokiniana* spore germination among the fungicides. MIC values were >100 $\mu$g/ml in both preparations. The most effective preparations for spore germination were Epoxiconazole+Pyraclostrobin (MIC:3 $\mu$g/ml) and Azoxystrobin+Difenocazole (MIC:3 $\mu$g/ml).

The efficacy (%) of the fungicides was calculated by measuring and comparing the diameters of mycelial growth of the causative agent in treated Petri dishes with those of control Petri dishes (Table 3) and Chlorotholanil and Chlorotholanil+Thiophonate-methyl were ineffective against *B. sorokiniana* isolate (Table 3) even at a dose of 100 $\mu$g/ml. Azoxystrobin+Difenocazole and Epoxiconazole+Pyraclostrobin presented with 100% inhibiton at doses of 3, 10 and 30 $\mu$g/ml while Trifloxystrobin was found to be 100% effective at doses of 10 and 30 $\mu$g/ml.

**Table 2** ED50 (μg/ml) and MIC (μg/ml) values of *Bipolaris sorokiniana* isolate against the applied fungicides.

| Fungicides | ED50 (μg/mL) | MIC (μg/mL) |
|---|---|---|
| Epoxiconazole+Pyraclostrobin | 0.04 | 3 |
| Metconazole | 10.3 | 30 |
| Chlorotholanil | 68 | >100 |
| Trifloxystrobin | 0.44 | 10 |
| Azoxystrobin+Difenocazole | 0.41 | 3 |
| Chlorotholanil+ Thiophonate-methyl | >100 | >100 |

**Table 3** The effectiveness of different doses of fungicides used in *in vitro* experiments on mycelial growth of *Bipolaris sorokiniana* isolate (%).

| Doses (μg/mL) | Effectiveness (%) | | | | | |
|---|---|---|---|---|---|---|
| | Epox.+Pyrac. | Metconazole | Chlorotholanil | Trifloxystrobin | Azoxy.+Dife. | Chlor.+Thiop.-methyl |
| 0.01 | 35.90 | 15.46 | – | 15.19 | 14.29 | – |
| 0.03 | 46.80 | 18.19 | – | 45.46 | 26.22 | – |
| 0.1 | 61.54 | 18.19 | – | 45.46 | 31 | – |
| 0.3 | 74.36 | 24.55 | – | 45.46 | 38.15 | – |
| 1 | 87.18 | 24.55 | 0 | 57.64 | 76.22 | 6.10 |
| 3 | 100 | 27.28 | 20 | 67.73 | 100 | 9.10 |
| 10 | 100 | 45.46 | 30 | 100 | 100 | 27.28 |
| 30 | 100 | 100 | 40 | 100 | 100 | 33.37 |
| 100 | – | – | 54 | – | – | 45.46 |

As a result, the most effective fungicides against *B. sorokiniana* were Azoxystrobin+Difenocazole and Epoxiconazole+Pyraclostrobin in Petri dish trials. İn the following studies, these fungicides were assessed under the greenhouse conditions.

## Efficacy of fungicide, plant activator, and their combinations against *B. sorokiniana* under greenhouse conditions

In the statistical analysis of the effects of fungicide and activator applications on *B. sorokiniana* in turfgrass, the (*L. acidophilus*-(Azoxystrobin+Difenoconazole recommended dose)) application was the most effective combination against *B. sorokiniana* with an effect value of 94.70%. The recommended dose application of the Azoxystrobin+Difenoconazole combination alone showed a high effect against the disease and ranked second with an effect value of 92.57% (Table 4). When the 1st sub-dose of the same fungicide was applied together with *L. acidophilus*, the effect was determined as 89.50%, and in its combination with the 2nd sub-dose, the effect was determined as 85.10% (Fig. 2). Other combinations that were examined in the study and had high efficacy values were Azoxystrobin+Difenoconazole 1st sub-dose, (*L. acidophilus*-(Azoxystrobin+Difenoconazole 1st sub-dose)), (*L. acidophilus*-(Epoxiconazole+Pyraclostrobin recommended dose)),

**Table 4  Effects values (%) of different concentrations of fungicides, activators and fungicide-activator combinations on *Bipolaris sorokiniana* in greenhouse condition.**

| Applications used in greenhouse trials against  *B. sorokiniana* | % Effect |
|---|---|
| (*Lactobacillus acidophilus*-(Azoxystrobin+Difenocazole recommended dose)) | $94.70 \pm 3.76^{*}$ |
| Azoxystrobin+Difenocazole recommended dose | $92.57 \pm 1.68^{ab}$ |
| Azoxystrobin+Difenocazole 1st subdose | $91.07 \pm 2.95^{abc}$ |
| (*Lactobacillus acidophilus*-(Azoxystrobin+Difenocazole 1st subdose)) | $89.50 \pm 2.78^{abc}$ |
| (*Lactobacillus acidophilus*-(Epoxiconazole+Pyraclostrobin recommended dose)) | $88.83 \pm 5.55^{abc}$ |
| (*Arthrobacter* sp.-(Epoxiconazole+Pyraclostrobin 1st subdose)) | $88.70 \pm 3.70^{abc}$ |
| (*Arthrobacter* sp.-(Azoxystrobin+Difenocazole recommended dose)) | $87.83 \pm 2.26^{abc}$ |
| Epoxiconazole+Pyraclostrobin recommended dose | $87.60 \pm 3.84^{abc}$ |
| (Harpin Protein-(Azoxystrobin+Difenocazole recommended dose)) | $86.20 \pm 3.37^{abcd}$ |
| (*Arthrobacter* sp.-(Epoxiconazole+Pyraclostrobin recommended dose)) | $86.07 \pm 8.56^{abcd}$ |
| (*Lactobacillus acidophilus*-(Azoxystrobin+Difenocazole 2nd subdose)) | $85.10 \pm 2.45^{bcde}$ |
| (*Arthrobacter* sp.-(Azoxystrobin+Difenocazole 1st subdose)) | $84.17 \pm 3.89^{bcde}$ |
| (*Lactobacillus acidophilus*-(Epoxiconazole+Pyraclostrobin 1st subdose)) | $83.93 \pm 1.99^{bcde}$ |
| Epoxiconazole+Pyraclostrobin 1st subdose | $83.53 \pm 6.15^{bcde}$ |
| (Harpin Protein-(Epoxiconazole+Pyraclostrobin recommended dose)) | $82.53 \pm 2.67^{cdef}$ |
| *Lactobacillus acidophilus* | $82.53 \pm 6.68^{cdef}$ |
| (Harpin Protein-(Azoxystrobin+Difenocazole 1st subdose)) | $82.27 \pm 4.20^{cdef}$ |
| (*Arthrobacter* sp.-(Azoxystrobin+Difenocazole 2nd subdose)) | $77.77 \pm 2.87^{defg}$ |
| (Harpin Protein-(Azoxystrobin+Difenocazole 2nd subdose)) | $76.83 \pm 5.30^{efg}$ |
| (*Arthrobacter* sp.-(Epoxiconazole+Pyraclostrobin 2nd subdose)) | $74.57 \pm 4.85^{fgh}$ |
| Azoxystrobin+Difenocazole 2nd subdose | $74.40 \pm 6.97^{fgh}$ |
| (Harpin Protein-(Epoxiconazole+Pyraclostrobin 1st subdose)) | $72.76 \pm 2.57^{gh}$ |
| Harpin Protein | $72.17 \pm 5.55^{gh}$ |
| Epoxiconazole+Pyraclostrobin 2nd subdose | $69.30 \pm 2.81^{ghi}$ |
| *Arthrobacter* sp. | $66.63 \pm 5.55^{hi}$ |
| (*Lactobacillus acidophilus*-(Epoxiconazole+Pyraclostrobin 2nd subdose)) | $62.10 \pm 2.74^{ij}$ |
| (Harpin Protein-(Epoxiconazole+Pyraclostrobin 2nd subdose)) | $56.47 \pm 10.19^{j}$ |
| ***P value*** | 0.001 |

**Notes.**

*Different lower cases indicate that significantly differences at $p < 0.001$ level.

(*Arthrobacter* sp.-(Epoxiconazole+Pyraclostrobin 1st sub-dose)), (*Arthrobacter* sp.-(Azoxystrobin+Difenoconazole recommended dose)), and Epoxiconazole+Pyraclostrobin recommended dose applications, which were in the same group with efficacy values of 91.07%, 89.50%, 88.83%, 88.70%, 87.83%, and 87.60%, respectively. No significant difference was found between them. After these combinations, the (Harpin Protein-(Azoxystrobin+Difenocazole 2nd subdose)) combination with 76.83% effectiveness, where the 2nd subdose of the fungicide was used, was remarkable (Fig. 2).

Among the fungicide combinations used in the study, the Epoxiconazole+Pyraclostrobin combination ranked second after the Azoxystrobin+Difenoconazole combination. While the Epoxiconazole+Pyraclostrobin combination showed an 87.60% efficacy value when used alone, the sub-dose of the same combination with *Arthrobacter* sp. showed an

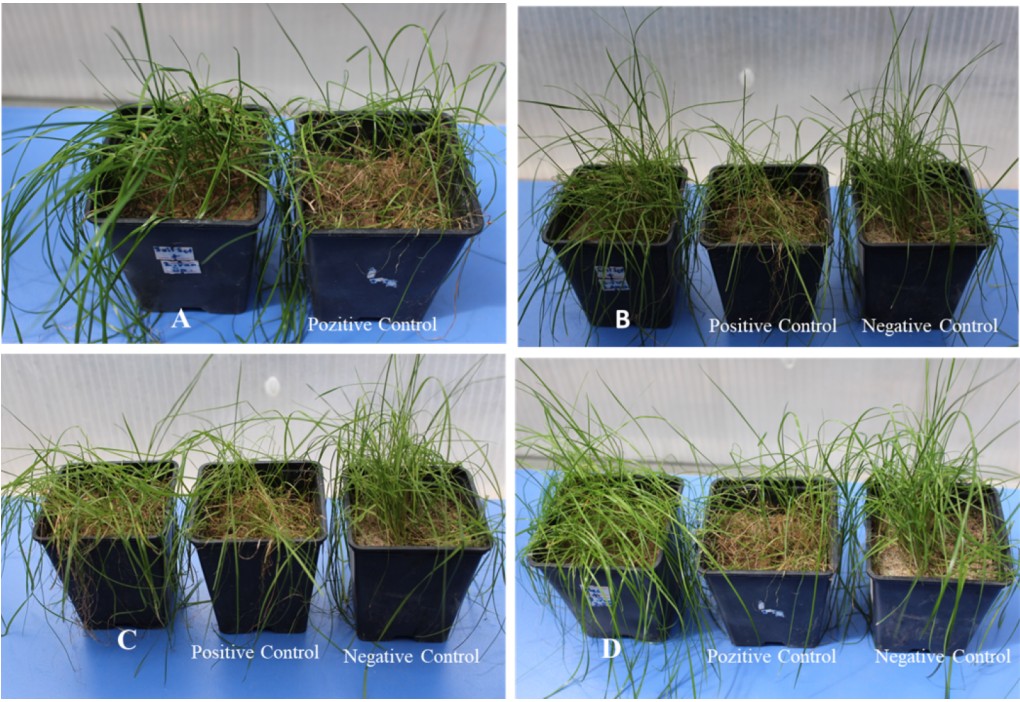

Figure 2 The greenhouse trials: (A) *L. acidophilus*-Azoxystrobin+Difenocazole recommended dose, (B) *L. acidophilus*-Azoxystrobin+Difenocazole 2nd subdose, (C) *Arthrobacter* sp. recommended dose, (D) Harpin-Azoxystrobin+Difenocazole 2nd subdose.

88.70% efficacy value. Although the efficacy results were statistically in the same group, it was evaluated as an important result because the lower dose of the fungicide was used. The 83.93% efficacy value obtained as a result of the application of (*L. acidophilus*-(Epoxiconazole+Pyraclostrobin 1st sub-dose)) is also important because the lower dose of the fungicide was used.

When the study was considered in terms of activators, *L. acidophilus* was the most effective activator. While *L. acidophilus* showed an 82.53% efficacy against the disease when used alone, the efficacy value increased with the recommended and lower doses of other fungicides. The results obtained from this activator especially with Azoxystrobin+Difenoconazole 1st and 2nd sub-dose (89.50% and 85.10) were significant since low dose fungicide was used. The effect values of harpin protein in all combinations with other fungicides remained below 87%. In the use of harpin protein alone, the effect was observed to be low at 72.17%. The highest combination with *Arthrobacter* sp. was the Epoxiconazole+Pyraclostrobin 1st sub-dose combination at 88.70% effect value. *Arthrobacter* sp. was the lowest application with a 66.63% effect value when used alone (Fig. 2). In the applications where *Arthrobacter* sp. was used together with different doses of fungicides, the effect values varied between 88.70% and 74.57% (Table 4).

As a result, it was decided to use the recommended dose of Azoxystrobin+Difenoconazole and *L. acidophilus*, which had the highest effect in the greenhouse trials, and additionally

the Azoxystrobin+Difenoconazole 1st sub-dose and *L. acidophilus* applications in field trials.

## Efficacy of fungicide, and plant activator combinations against *B. sorokiniana* in area conditions

The studies were carried out in two locations: Ankara and Antalya provinces, with the combination of (*L. acidophilus*-(Azoxystrobin+Difenocazole recommended dose)) and (*L. acidophilus*-(Azoxystrobin+Difenocazole 1st subdose)) tested against *B. sorokiniana*. The average efficacy of the combination of (*L. acidophilus*-(Azoxystrobin+Difenocazole recommended dose)) was 93.27% and was 89.90% for (*L. acidophilus*-(Azoxystrobin+Difenocazole 1st subdose)) in the trials for *B. sorokiniana* in Ankara (Table 5). A statistically significant difference was not observed between these two results. Similar results were obtained for these combinations in Antalya, there was also no statistical difference between the results (Table 5). As a result, it was concluded that instead of the recommended dose of fungicides, lower doses of this chemical can be used alternatively with activators in *B. sorokiniana* control programs. Several studies around the world have determined that activators increase the effect of fungicides (*Quimette, 2012*; *Ons et al., 2020*). Supportingly, it was also observed that the activator used in the study increased the effect levels of fungicides.

## DISCUSSION

In this study, three plant activators (*Arthrobacter* sp., *L. acidophilus* and harpin protein) and two fungicides (which were found to be effective in *in vitro* studies) against *B. sorokiniana*, which is a problem in the turfgrass, were used. The effects of using different doses of Epoxiconazole+Pyraclostrobin and Azoxystrobin+Difenocazole alone as fungicide and their alternate usage with three activators were investigated and an environmentally friendly control program was tried to be created. In the greenhouse studies, the effects of (*L. acidophilus*-(Azoxystrobin+Difenocazole recommended dose)) and (*L. acidophilus*-(Azoxystrobin+Difenocazole 1st subdose)) applications, which were found to be the most effective, against *B. sorokiniana,* also were investigated under area conditions. As a result, it was concluded that the alternating application program in which a lower dose of fungicide with an activator when applied, it can be effective for environmentally friendly programs in the control of diseases. It is stated that these types of applications increase fungicide effectiveness and delay resistance to fungicides, as well as being environmentally friendly (*Nobutaka et al., 2006*). There are many studies in which plant activators are used together with fungicides to fight fungal diseases in different hosts. In these studies, it is understood that fungicide and activator combinations give more effective results (*Tosun et al., 2003*; *Boyraz, Kaymak & Baştaş, 2006*; *Dereboylu & Tort, 2010*; *Tosun & Turan, 2011*). In three studies were done in the USA, some bacterial strains isolated from the green parts of turfgrass plants were found to be effective in controlling some leaf, root and root collar diseases in turfgrass through systemic resistance (*Giesler & Yuen, 1998*; *Zhang & Yuen, 2000*; *Kilic-Ekici & Yuen, 2004*). For example, *Lysobacter enzymogenes* C3 strain was found to be effective in reducing the severity of necrosis by *Rhizotonia solani*, which causes

**Table 5** Effects of fungicide-activator combinations on *Bipolaris sorokiniana* in field trials conducted in Ankara and Antalya provinces.

| Location | Application | Mean | Coefficient of variation |
|---|---|---|---|
| Ankara | (*Lactobacillus acidophilus*-(Azoxystrobin+Difenocazole recommended dose)) | 93.27 ± 4.02[*] | 4.312 |
| | (*Lactobacillus acidophilus*-(Azoxystrobin+Difenocazole 1st subdose)) | 89.90 ± 2.99 | 3.334 |
| *P value* | | 0.310 | |
| Antalya | (*Lactobacillus acidophilus*-(Azoxystrobin+Difenocazole recommended dose)) | 95.00 ± 2.92[*] | 3.074 |
| | (*Lactobacillus acidophilus*-(Azoxystrobin+Difenocazole 1st subdose)) | 91.17 ± 2.87 | 3.145 |
| *P value* | | 0.180 | |

Notes.

[*]n.s.; There are not significantly differences between the two applications in both of the two provinces separately at least $p < 0.05$ level.

brown patch disease, as well as *B. sorokiniana*, which causes leaf spot disease, in *Festuca arundinacea* grass variety (*Giesler & Yuen, 1998*; *Kilic-Ekici & Yuen, 2004*). In another study, *Tosun & Turan (2011)* investigated the effectiveness of control programs consisting of plant activators, and biological and effective fungicides against *R. solani*, which causes root and crown root disease in grasses. As a result of the applications, ((*L. acidophilus* fermented product + tolclofos methyl+thiram) + trifloxystrobin) and (*Streptomyces lydicus* strain WYEC 108 + azoxystrobin) showed the best results.

There are a few studies on the effectiveness of harpin protein on disease development except turfgrass. It has been successfully applied and satisfactory results have been obtained against many disease pathogens such as *Penicillium expansum* (blue mold) on apples, *Alternaria solani* (early leaf blight) on tomatoes and canola, *Verticillium dahlia* and *Botrytis cinerea* on peppers, *Phytophthora infestans* on tomatoes, anthracnose on cucumbers, and *Cercospora coffeicola* (leaf spot) on coffee plants (*Strobel et al., 1996*; *De Capdeville et al., 2003*; *Bishnoi & Payyavula, 2004*; *Akbudak et al., 2006*; *Tezcan & Akbudak, 2009*).

To evaluate methods of controlling apple black spot disease (*Venturia inaequalis*) in Golden variety apples, some plant activators and fungicides were applied alone or in combinations three times during the early stages of plant development. According to the data obtained, the first two applications ISR-2000 (*L. acidiophilus* fermentation product + yeast and plant extract + benzoic acid) and the last application ISR-2000+ Chorus exhibited the highest efficiency of 73.10%. This was followed by the combination of ISR-2000+Candit with a rate of 67.81%. In fungicides, Chorus alone (58.77%) and Candit alone (55.74%) were moderately efficacious when applied three times. The use of İSR-2000 alone, compared to Crop-set (*L. acidiophilus* fermentation product + plant extract + mineral substance) showed a higher effect against apple black spot disease. The crop-set application alone has been shown to promote disease compared to control (*Boyraz, Kaymak & Baştaş, 2006*). Successful results have been obtained in the control of some diseases on tomatoes (*Tosun et al., 2003*), pepper (*Konukoğlu, 2007*), and cucumber (*Dereboylu & Tort, 2010*) using plant activators alone and their combination with fungicides in Türkiye. Studies have shown that combinations of plant activators and fungicides are more effective in fungal disease control. For instance, in a study, it was reported that the application of a combination of plant activator and fungicide against *P. infestans* on potatoes provided significantly increased potato quality, and yield in addition to the controlling of the disease (*Özdemir, 2023*). The

results of this study are in parallel with the results of many researchers that increasing plant resistance against plant fungal diseases with plants activators (*Tosun et al., 2003*; *Boyraz, Kaymak & Baştaş, 2006*; *LaMondia, 2009*; *Dereboylu & Tort, 2010*; *Tosun & Turan, 2011*; *Ingle et al., 2014*; *Delisoy & Altinok, 2019*). Similarly, it was released that the combination of fungicide and plant activator was found more effective against bacterial plant diseases when applied together than when applied alone (*Karabay et al., 2003*; *Türküsay & Tosun, 2005*; *Ustun, Demir & Saygili, 2005*).

A combination of *Pseudomonas fluorescens*, *Mesorhizobium cicero*, and *T. harzianum* with the fungicide carboxin and thiram provided the highest seed germination, grain yield, and the lowest wilt incidence caused by *F. oxysporum* in pot and field experiments of chickpea (*Dubey et al., 2015*). The combination of *T. harzianum*, *P. fluorescens*, and carbendazim was more effective against *Magnaporthe oryzae* in comparison to their application in field experiments of rice (*Jambhulkar et al., 2018*).

The effects of plant activators on crop yield have also been demonstrated by various studies. The applications of plant activators alone or in combination with fungicides such as cyprodinil and fenpropidin provided better product development in wheat. In addition, the regular application of plant activators causes an increasing in production. Plant activator application applied alone yielded a 9% increase in wheat production, this rate was 13% when plant was used with fenpropidin, and 17% with cyprodinil mixtures. After the application of plant activator in wheat, newly developing plant parts were less susceptible to disease and were therefore healthier. The studies in wheat have shown that activators are not only effective on *Blumeria graminis* but are also effective against *Septoria* sp., *Puccinia recondita*, *Pseudocercosporella herpotrichoides*, and *B. sorokiniana* (*Aminuzzaman & Hossain, 2007*). In a study conducted on red peppers, 1,200 kg of fruit was harvested per decare as disease-free product, while 2,088 kg of product per decare was obtained from the plots on which CropSet was applied. 2,448 kg of red pepper was harvested from the ISR 2,000 plots (*Karavaş, 2002*).

## CONCLUSIONS

Due to the increasing fungal complaints in the expanding turfgrass areas, especially in golf courses, too many fungicides have been used. The increase in the use of fungicides in turfgrass areas, which also have very high maintenance costs, increases the cost and causes environmental pollution. However, their widespread use leads to the emergence of pathogens resistant to fungicides (*Chawla et al., 2018*). Biological activators are recommended as an alternative to standard fungicides in agricultural production, but when used alone, their disease management capacity is generally insufficient. This is largely due to uncontrolled environmental conditions (*Ons et al., 2020*). It has been determined that the use of fungicides together with biological preparations increases their effectiveness against diseases compared to their use alone (*Nobutaka et al., 2006*). For these reasons, the main purpose of this study was to present an integrated approach that combines biological and microbial activators with fungicides and thus to find a way to reduce fungicide doses to manage plant diseases. Another desired outcome of this study is the strategy of combining

control mechanisms with different active ingredients and modes of action, reducing the selection pressure on pathogens and thus the chance of developing resistance. However, to allow for large-scale application, more information is needed, including the timing of repeated applications, the number and interval of applications, and compatibility with other fungicides in graminea and other crop plants other than turfgrass. The compatibility of activators with fungicides may differ when used in mixtures or rotationally. These studies should be compared separately.

### Funding
This study was supported by the Technological Research Council of Türkiye (TÜBİTAK) (Project No: 114O400). The funders had no role in study design, data collection and analysis, decision to publish, or preparation of the manuscript.

### Grant Disclosures
The following grant information was disclosed by the authors:
The Technological Research Council of Türkiye (TÜBİTAK) (Project No: 114O400).

### Competing Interests
The authors declare there are no competing interests.

### Author Contributions
- Filiz Ünal conceived and designed the experiments, performed the experiments, analyzed the data, prepared figures and/or tables, authored or reviewed drafts of the article, and approved the final draft.
- İlker Kurbetli conceived and designed the experiments, performed the experiments, authored or reviewed drafts of the article, and approved the final draft.
- Yeşim Eğerci conceived and designed the experiments, performed the experiments, prepared figures and/or tables, and approved the final draft.
- Aysun Cavusoglu analyzed the data, prepared figures and/or tables, authored or reviewed drafts of the article, and approved the final draft.

### Data Availability
The raw data is available in the Supplemental Files.

### Supplemental Information
Supplemental information for this article can be found online at http://dx.doi.org/10.7717/peerj.18943#supplemental-information.

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
