# Peer review of "Effects of chemical fungicides combined with plant resistance inducers against Bipolaris sorokiniana in turfgrass"

_PeerJ, doi:10.7717/peerj.18943_

## Round 0.1 · original submission · Major Revisions

Dear Authors

The reviewers have recommended revisions to your manuscript. Therefore, I invite you to respond to the reviewers' comments and revise your manuscript.
In addition, there are significant concerns about the manuscript's grammar, usage, and overall readability. We, therefore, request that you revise the text to fix the grammatical errors and improve the overall readability of the text.

With Thanks

Reviewer 1 ·

Basic reporting

The article titled "Effects of Chemical Fungicides Combined with Plant Resistance Inducers Against Bipolaris sorokiniana in Turfgrass" requires a thorough revision to improve clarity, coherence, and logical progression. The language contains numerous typos, grammatical, structural issues and awkward rephrasing. Simplify sentences, avoid redundancy, and ensure proper punctuation. Perform a thorough grammar and spell check.
Please refine the keywords section. It is important to provide 5-7 distinct keywords that are not repeated from the title.
The introduction is a bit disjointed, jumping from one topic to another without clear transitions. For example, after discussing the importance of turfgrass areas and the issue of fungal diseases, the text abruptly moves to the development of resistance to fungicides without properly connecting the two. Use clearer transitions between different sections. For instance, explain how the increase in turfgrass areas leads to overuse of fungicides, which in turn causes resistance, before moving on to biological solutions.
The specific research gap is not clearly defined. It mentions biological alternatives, but it is unclear why these alternatives were chosen and what makes this study novel. Highlight the specific gap that this study aims to address—why is this approach using Lactobacillus acidophilus and Harpin proteins important compared to previous methods? Clarify what the study contributes to existing knowledge. Focus more on establishing the problem and how this study proposes to solve it. The references are inconsistently integrated into the narrative. Some claims (e.g., the increase in turfgrass areas) are unsupported by evidence. Ensure that all major claims, especially those that form the basis of the study, are backed by relevant and recent literature.
The methodology focuses heavily on mycelial growth inhibition but provides limited information about how spore germination inhibition was assessed. Provide a clear and detailed protocol for the spore germination assay, including the incubation times, media used, and how spore germination was visually or microscopically assessed (Please provide photographs or micrographs). Mention the criteria for determining spore viability. Similarly, the description of how mycelial growth was measured is unclear. It mentions diameter measurements, but it is not clear whether measurements were taken after a set incubation period or at multiple time points. Similarly, the method for calculating ED50 values, while referenced, could benefit from a clearer explanation. Provide a step-by-step breakdown of how the measurements were taken (e.g., after how many days post-inoculation) and explain the ED50 calculation method in simpler terms, even if a reference is provided. Mention the software or statistical method used for log-probit analysis to improve transparency and reproducibility. Explain what the control treatment consists of (e.g., solvent-only or untreated PDA). Include any additional control conditions, such as untreated fungus grown in the absence of fungicides, to assess the baseline growth rate. Elaborate on how the randomization was implemented (e.g., random assignment of Petri dishes or treatments) and whether this randomization was done for each repetition or across all experimental setups. Provide more information on the environmental conditions maintained during the in vitro trials (e.g., light/dark cycles or relative humidity). If these factors were controlled or not deemed relevant, it should be mentioned for clarity.
The descriptions of the experimental procedures, especially for fungicide and activator application, lack sufficient detail. For instance, “sequential application of activators and fungicides” is mentioned, but the exact timing, methods of application, and intervals between treatments need to be explained more explicitly. This can lead to difficulties in replication. The methodology involves spraying fungicides and activators onto 15-day-old plants and then inoculating with the pathogen. While this provides an opportunity for preventive control, there is no mention of why this specific timing was chosen or whether curative effects were also tested. Explain why applications were performed at this stage of plant development and whether both preventive and curative effects of the fungicides and activators were assessed.
Avoid repeating the same citations unless introducing new information from those studies.
When describing the biological control agents (e.g., Lactobacillus acidophilus, Arthrobacter sp., and Harpin protein), there is a lack of detail on the strains used or how they were sourced and prepared for the experiments. Explain the source of the activators (e.g., commercially available or lab-cultured) and preparation methods.
Discuss the potential effects of soil sterilization on the experiment, particularly in relation to plant-microbe interactions. Autoclaving soil for 45 minutes at 121°C can lead to the breakdown of organic matter, and the use of non-sterile but pathogen-free soil may provide a more realistic growing condition.
The post-inoculation period is managed by placing the pots in moistened polyethylene bags to maintain 95-100% relative humidity for 72 hours. This technique is commonly used to enhance infection, but the method could lead to an artificial microclimate that doesn't reflect field conditions. Consider whether the use of polyethylene bags is necessary and whether a more field-realistic humidity management approach could be used. If this step is retained, provide more details about how uniform humidity was ensured across replicates.
The 0-5 scale used to assess disease severity is standard in plant pathology, but it could be somewhat subjective, particularly when distinguishing between levels of necrotic spots and yellowing. Use digital imaging tools or software (e.g., ImageJ) to quantify disease severity more objectively, especially in large-scale studies. This would reduce observer bias and improve the reproducibility of results.
The field trials follow a similar methodology to the greenhouse studies, with artificial inoculation being performed at two different sites. However, no details are provided on how the environmental conditions at these two locations (Ankara and Antalya) may differ and how this might affect the pathogen’s development. Include information on environmental factors (e.g., temperature, humidity, soil type) at the two field sites. This is crucial for understanding whether results are site-specific or generalizable. Additionally, discuss how inoculation consistency across the sites was ensured
Re-isolation of Bipolaris sorokiniana from infected leaves is a critical step in confirming that the observed disease symptoms are indeed due to the inoculated pathogen. However, there are no details on how re-isolation was performed or whether molecular techniques were used to confirm the identity of the re-isolated pathogen. Provide a clear protocol for re-isolation and confirmation of the pathogen’s identity. Consider using molecular markers to ensure that the re-isolated fungus is genetically identical to the original inoculum.
The results section lacks depth and appears to merely restate the data shown in the tables without sufficient interpretation. To improve, the results should be presented in a more scientific manner, incorporating detailed discussions of the findings. This includes highlighting statistically significant differences among the treatments and analyzing the percentage increases or decreases in ED50 and MIC values with respect to varying doses of the fungicides. Moreover, comparisons should be drawn to demonstrate trends, such as which fungicide treatments show the most effective reductions in disease incidence or pathogen inhibition, backed by proper statistical tests to validate the findings.
In discussion, while numerous studies are referenced, the authors do not deeply delve into how the mechanisms of action of the activators differ or complement the fungicides. For instance, Lactobacillus acidophilus, Arthrobacter sp., and harpin protein likely function through distinct biochemical pathways, but these are not clearly explained in the discussion. An expert reader might expect a more mechanistic exploration. Although the use of activators is well-documented, the study lacks novel insights into their specific interaction with fungicides against B. sorokiniana beyond what has already been reported. The parallels drawn to previous work suggest that while the study corroborates earlier findings, it doesn’t push the boundaries of current knowledge. The discussion is heavily reliant on referencing earlier studies without critically engaging with the limitations or variations of the current work. There is little analysis of how variations in environmental conditions, turfgrass species, or pathogen variability could influence the efficacy of the treatments. The data interpretation seems overly simplistic. The discussion should have explored potential reasons for the differential efficacy of fungicide-activator combinations, especially the inconsistencies in effectiveness between the different activators (e.g., Arthrobacter sp. vs. Lactobacillus acidophilus). Also, the lack of significant differences between certain combinations should have been critically assessed.
The conclusion does not offer enough practical recommendations for turfgrass managers or researchers. For example, how should these findings influence fungicide application protocols in real-world settings? There’s a missed opportunity to provide actionable insights based on the study’s results. The conclusion could have been expanded to discuss how these findings may apply to other fungal pathogens in turfgrass or even other crops. Without this broader context, the significance of the study is somewhat constrained to B. sorokiniana in turfgrass. The conclusion doesn’t adequately address the limitations of the study. For instance, greenhouse and in vitro trials are not always directly translatable to field conditions. It would have strengthened the paper to acknowledge these limitations and suggest specific areas where further research is needed to confirm the efficacy of these treatments in real-world settings.
Specific comments:
“The turfgrass areas which also require continuous irrigation, which causes problems related diseases have been increased,” is grammatically incorrect. A more precise version would be, “Turfgrass areas that require continuous irrigation have increased, leading to disease-related problems.”
The usage of "1t" instead of "It" appears throughout the text, which is a typographical error.
The use of "1n" instead of "In" and "1rrigation" instead of "Irrigation" are recurring typographical errors, which may affect the readability and professionalism of the manuscript.
The article contains redundant use of the word "which" in multiple instances, creating unnecessarily long and confusing sentences. This needs to be simplified for better readability.
Phrases like (Lines 43) "day by day turfgrass areas... are increasing" are vague. While the sentiment is understandable, such statements need to be backed with data or references to substantiate the claim. Additionally, the environmental challenges of global warming are mentioned, but they are not well-connected to turfgrass management. Provide specific examples or data to support claims, such as recent statistics on the increase in turfgrass areas. If mentioning global warming, connect it explicitly to irrigation or disease pressures in turfgrass areas.
Provide a clearer definition of what "virulent (98%)" means. Specify the assay or metric used to determine this virulence level and clarify the significance of this figure in relation to other isolates.
To enhance the scientific rigor of the study, additional visual data in the form of graphs and micrographs, including ED50 (µg/ml) and MIC (µg/ml) values for all concentrations of fungicides tested, should be included. This would provide clearer, more detailed evidence of the fungicidal efficacy against Bipolaris sorokiniana isolates. Such visual aids would illustrate the trends and variations in disease incidence and the pathogen's response to different fungicides, allowing for a more comprehensive analysis of the fungicides' inhibitory effects, supported by clear statistical comparisons.
Disease incidence should be provided in both greenhouse studies and field trials.
How did you calculated effectiveness of of different doses of fungicides in (as given in table 3) in vitro experiments on mycelial growth of Bipolaris sorokiniana isolate?
In some areas, sentences are unnecessarily complex or long. For example, “The spore suspension was adjusted to 1x105 spores/ml using a Thoma slide” could be rephrased as “The spore concentration was adjusted to 1x105spores/ml using a Thoma counting chamber.”
(Lines 122-125): The effectiveness of the fungicides (Epoxiconazole+Pyraclostrobin, Metconazole, Chlorotholanil, Trifloxystrobin, Azoxystrobin+Difenocazole, Chlorotholanil+Thiophonate-methyl) against mycelial growth and spore germination of the fungus was determined. How?
(Line 125-126)1n addition, the most effective fungicide doses and the sensitivity levels of the pathogen to the fungicides were determined.
(Lines 130-132) Stock solutions were prepared higher doses from which dilutions were made to obtain the desired fungicide doses of 10000, 1000, 100 ppm E.S doses. Clarify the sentence. Mention the exact concentration of fungicide in stock solution.
(Line 138) “Experiments were set up in a randomized plot design in triplic.” Randomized block design?
(Lines 159-162) 1n greenhouse studies, sterile garden soil (autoclaved for 45 minutes at 121°C), burnt farm manure and fine sand (2:1:1) was mixed and filled in 10x10x11 size and 0,8 L volume sterile plastic pots. Clarify the procedure and remove typos and grammatical errors.
(Lines 263-264) Several studies around the world have determined that activators increase the effect of fungicides (Quimette, 2012). Use recent citations for such claims.
Lines 282-283 (Tosun et al., 2003; Boyraz et al., 2006; Dereboylu and Tort, 2010; Tosun and Turan, 2011): Limit the number of citations to a maximum of three throughout the manuscript, prioritizing the most recent and relevant references.
The caption of Table 4 and 5, "Effect values (%) obtained as a result of trials with Bipolaris sorokiniana," needs grammatical correction and detailed clarification. Additionally, indicate that data are presented as mean ± standard deviation (SD).
In Table 4 and 5, as well as in multiple sections of the manuscript, commas are incorrectly used in place of decimal points. Please correct this typographical error throughout the manuscript to ensure proper formatting and consistency in numerical data representation.
How values in Table 5 were calculated?

Experimental design

The experimental design in the current manuscript is insufficiently detailed and requires substantial revision. Although a randomized block trial is mentioned for the field study, the authors must clearly and comprehensively describe the experimental setup. Additionally, the manuscript lacks effective presentation of crucial data, particularly in regard to disease incidence in both control and treated plots. The absence of this data severely undermines the ability to assess the efficacy of the proposed treatments, calling into question the validity and robustness of the study. It is essential to present clear, well-organized data to substantiate the conclusions drawn from the experiments.

Validity of the findings

The manuscript is missing essential data, particularly with respect to plant health and disease incidence, which seriously compromises the validity of the findings. To ensure a scientifically sound study, it is imperative that clear, quantitative data be presented, highlighting the differences in disease incidence between control and treated plots. Without this data, it becomes challenging to accurately evaluate the efficacy of the fungicides and activators, and to validate the conclusions drawn from the study. The authors are strongly encouraged to include comprehensive, statistically analyzed data that supports their claims, with a detailed comparison of both control and treatment outcomes presented in a meaningful and scientifically rigorous manner.

Reviewer 2 ·

Basic reporting

The article is suitable for publication in terms of format and language. Reducing the use of fungicides is of great importance for environmental protection. However, it is recommended to add sources from the last 5 years in the introduction to emphasize the importance and currency of the subject.

Experimental design

no comments

Validity of the findings

no comments

Additional comments

.

Reviewer 3 ·

Basic reporting

The article was read and reviewed. Suggestions and corrections are made in the text of the article. Unfortunately, there are many writing and grammar problems in the text of the article, which should be checked by a native person who is fluent in English.
It is better to include photos of the pathogenicity and effectiveness of the compounds used in the article.

Experimental design

The article was read and reviewed. Suggestions and corrections are made in the text of the article. Unfortunately, there are many writing and grammar problems in the text of the article, which should be checked by a native person who is fluent in English.
It is better to include photos of the pathogenicity and effectiveness of the compounds used in the article.

Validity of the findings

The article was read and reviewed. Suggestions and corrections are made in the text of the article. Unfortunately, there are many writing and grammar problems in the text of the article, which should be checked by a native person who is fluent in English.
It is better to include photos of the pathogenicity and effectiveness of the compounds used in the article.

Additional comments

The article was read and reviewed. Suggestions and corrections are made in the text of the article. Unfortunately, there are many writing and grammar problems in the text of the article, which should be checked by a native person who is fluent in English.
It is better to include photos of the pathogenicity and effectiveness of the compounds used in the article.

Annotated reviews are not available for download in order to protect the identity of reviewers who chose to remain anonymous.

---

## Round 0.2 · Minor Revisions

Dear Authors
The manuscript still needs a minor revision before publication. The authors are invited to revise the paper considering all the suggestions made by the reviewers. Please note that the requested changes are required for publication.
With Thanks

Reviewer 1 ·

Basic reporting

The authors address the majority of the queries raised in the previous round of revision. However, there are still a few remaining issues that require further attention.

Typographical mistakes still exist in the manuscript, like in: Line 53: "Bipolaris sp., which was previously called previously Helminthosporium" Remove the repeated word "previously." Line 75 "keep polluting the environment and groundwater resources", replace "keep polluting" with "continue to pollute." Line 76 "spreaded in nature", replace "spreaded" with "spread." Line 126"activates the plant resistance mechanism, decomposes pesticides, fixes nitrogen, produces beneficial enzymes, sewage treatment", Add "and" before "sewage treatment" to correct the list format. Line 135"chemical pesticides on the environment are reduced, the risk of resistance to fungicides". Replace the comma with "and" for grammatical accuracy. Carefully check material and methods, results, discussion and conclusion sections for such refinements.

In the materials and methods section, properly add equations using equation mode.

Add subheadings in the materials and methods section.

The Townsend and Heuberger formula and Abbott’s formula should be appropriately placed following the description of the disease assessment. Additionally, the statistical analysis should be discussed under a separate subheading to ensure clarity and facilitate reproducibility.

Line 318-319 The statement “In the applications where Arthrobacter sp. was used together with different doses of fungicides, the effect values varied between 88,70%-74,57%” is ambiguous and lacks clarity. Please rephrase this result to explicitly mention the specific doses of fungicides used, along with the corresponding effect values, to provide a clearer understanding of the observed variations.

Please round off all percentage values to whole numbers for consistency and clarity. For example, 76.83% should be rounded to 77%. Ensure this adjustment is applied uniformly throughout the manuscript.

Remove all outdated references from the introduction and discussion sections, like Giesler and Yuen, 1998.

Please remove references from the conclusion section. Conclude your findings appropriately by clearly highlighting the key results, practical implications, and recommendations derived from the study.
Focus on presenting a concise and impactful summary that emphasizes the relevance of the integrated use of fungicides and biological activators without referencing external sources.

Experimental design

Experimental design seems appropriate.

Validity of the findings

Findings seems valid.

Reviewer 2 ·

Basic reporting

The requested corrections were added to the article by the author.

Experimental design

no comment

Validity of the findings

no comment

---

## Round 0.3 · Minor Revisions

Dear Authors

The manuscript still needs a minor revision before publication. The authors are invited to revise the paper considering all the suggestions made by the reviewers. Please note that the requested changes are required for publication.
Best Regards

Reviewer 1 ·

Basic reporting

I am pleased to see authors have tried to address the queries raised in the second round of revision. But still the article needs careful revision of the following:
The introduction is lengthy, overly detailed, and lacks a clear logical flow. Certain points are repeated, which affects readability. For example: Repeated intensive and incorrect fungicide applications (lines 72-77). Reduction in the amount and frequency of chemicals (lines 79-81). Please combine similar ideas and streamline redundant information for conciseness. Extensive descriptions of chemical fungicides and biological control mechanisms (lines 83–131) could be shortened. Summarize well-known facts like harpin protein and Trichoderma antagonism while emphasizing their relevance to the current study. Maintain focus on turfgrass-specific disease management strategies.
Multiple references are provided without sufficient explanation of their relevance (e.g., Nobutaka, 2006; Thakur and Sohal, 2013 in lines 83–88) and (Keller et al., 2000; de Capdeville et al., 2003; Bishnoi and Pavyavula, 2004; Akbudak et al., 2006; Tezcan and Akbudak, 2009). Briefly indicate the relevance or findings of these studies, rather than listing them in bulk or reduce the number of intext citations by keeping most relevant and latest three references.

Typographical errors like " continue to pollute keep polluting the environment" (line 75) and inconsistencies in verb tense are noticed at multiple places.
Citations (e.g., Shah et al., 2006; Anonymous, 2024) (Line 153) are vague, and the use of "Anonymous, 2024" raises concerns about the reliability of the reference. Ensure proper attribution and context for each cited study.
The results lack a clear and concise summary of the findings. While data such as ED50, MIC, and efficacy values are provided, the narrative is overly detailed and repetitive. Present key findings in a more structured and succinct manner.
Statements like (Lines 282-320) “statistically in a different group” are vague and should be supported by specific p-values or test results and (Lines 323-335), “lower doses of this chemical can be used alternatively with activators” is compelling but lacks sufficient statistical evidence. Provide statistical metrics such as confidence intervals or p-values alongside comparisons. Make such corrections throughout the results section.

Experimental design

Experimental design seems appropriate.

Validity of the findings

Findings seems promising.

Reviewer 3 ·

Basic reporting

appropriate

Experimental design

appropriate

Validity of the findings

very interesting

---

## Round 0.4 · accepted · Accept

Dear Authors,

I am pleased to inform you that the manuscript can be accepted for publication.
Congratulations on accepting your manuscript and thank you for your interest in submitting your work to PeerJ.

With Thanks

Reviewer 1 ·

Basic reporting

The authors have addressed most of the suggested revisions. The manuscript now meets the necessary standards and is recommended for acceptance for publication

Experimental design

Experimental design seems reproducible.

Validity of the findings

The findings appear to be generally valid and align with the data presented.